# Platelet Receptor Activity for Predicting Survival in Patients with Intracranial Bleeding

**DOI:** 10.3390/jcm10102205

**Published:** 2021-05-19

**Authors:** Barbara Dragan, Barbara Adamik, Malgorzata Burzynska, Szymon Lukasz Dragan, Waldemar Gozdzik

**Affiliations:** 1Department of Anaesthesiology and Intensive Therapy, Wroclaw Medical University, Borowska 213, 50-556 Wroclaw, Poland; barbara.adamik@umed.wroc.pl (B.A.); malgorzata.burzynska@umed.wroc.pl (M.B.); waldemar.gozdzik@umed.wroc.pl (W.G.); 2Department of Regenerative and Restorative Medicine in Orthopaedics, Wroclaw Medical University, Borowska 213, 50-556 Wroclaw, Poland; s.dragan@umed.wroc.pl

**Keywords:** platelet aggregation, intracranial hemorrhages, craniocerebral trauma, aneurysm, ruptured, platelet function tests, survival rate

## Abstract

Blood coagulation disorders in patients with intracranial bleeding as a result of head injuries or ruptured aneurysms are a diagnostic and therapeutic problem and appropriate assessments are needed to limit CNS damage and to implement preventive measures. The aim of the study was to monitor changes in platelet aggregation and to assess the importance of platelet dysfunction for predicting survival. Platelet receptor function analysis was performed using the agonists arachidonic acid (ASPI), adenosine diphosphate (ADP), collagen (COL), thrombin receptor activating protein (TRAP), ristocetin (RISTO) upon admission to the ICU and on days 2, 3, and 5. On admission, the ASPI, ADP, COL, TRAP, and RISTO tests indicated there was reduced platelet aggregation, despite there being a normal platelet count. In ‘Non-survivors’, the platelet response to all agonists was suppressed throughout the study period, while in ‘Survivors’ it improved. Measuring platelet function in ICU patients with intracranial bleeding is a strong predictor related to outcome: patients with impaired platelet aggregation had a lower 28-day survival rate compared to patients with normal platelet aggregation (log-rank test *p* = 0.014). The results indicated that measuring platelet aggregation can be helpful in the early detection, diagnosis, and treatment of bleeding disorders.

## 1. Introduction

Blood coagulation disorders in patients with acute intracranial bleeding are a common diagnostic and therapeutic problem in neurocritical care departments [1]. Intracranial bleeding from head injuries and ruptured aneurysms are associated with high morbidity and mortality, even in young people [2,3,4]. Those who survive the initial bleeding, often suffer cognitive decline and functional impairment, which in turn results in the deterioration of the patient’s quality of life and represents a serious public health problem [5,6,7]. Appropriate assessments are needed for the timely and accurate prediction of the development and severity of intracranial hemorrhage-induced coagulation disorders to limit CNS damage and to implement preventive measures and targeted therapies. While the importance of abnormal routine coagulation tests following head injuries and subarachnoid hemorrhages is well known, the complex pathophysiological mechanisms of coagulopathies and the role of platelets in these disorders are not well known and require further investigation [8].

More than one-third of patients with acute CNS hemorrhagic damage from traumatic brain injury (TBI) have abnormal coagulation test results on admission, and the results are linked to a progression of hemorrhagic cerebral lesions and a higher mortality rate [9,10,11]. Previous studies have found decreased platelet function in TBI patients and also identified platelet dysfunction as an independent predictor of mortality in trauma patients [12,13]. Inhibition of the adenosine diphosphate (ADP) receptor and the arachidonic acid (AA) receptor on platelets was an important feature of TBI in a human and animal model of coagulation disturbances and was associated with the severity of brain damage [14]. A later study only partially confirmed these results: patients with TBI showed a significantly lower platelet response to an agonist, arachidonic acid (AA), suggesting the possible involvement of the cyclooxygenase pathway in platelet dysfunction, even though the platelet response to ADP was not different from that of healthy controls [12]. So far, little attention has been paid to platelet activation in patients with acute SAH-induced CNS hemorrhagic injury, but some studies have shown significant changes in aggregation. Perez et al. demonstrated increased activation and aggregation of platelets in SAH patients, especially after stimulation with ADP [15]. Different results were presented by Frontera et al., who showed that the mean activation of platelets remained within the reference range, regardless of the severity of SAH according to the Hunt–Hess scale [16] and by He et al., who demonstrated an inhibition of platelet activation in response to ADP and AA [17]. These ambiguous or even contradictory results indicate that in order to comprehensively determine hemostatic disorders in patients with intracranial bleeding, there should be a supplementary assessment of platelet function.

Platelet function is not routinely measured in many neurocritical care departments, but with the introduction of point-of-care whole-blood platelet function tests, it can now be easily assessed [18]. There are now diagnostic measuring devices that a doctor or nurse can run in an operating room or intensive care unit, and the results are available in minutes. The total hands-on time is very short, only a few minutes, and a whole-blood sample is used for testing, so there is no need to pre-process the sample. The aim of this study was to monitor changes in platelet aggregation in patients with acute intracranial bleeding as a result of head injuries or ruptured aneurysms, and to assess the importance of platelet dysfunction for predicting survival.

## 2. Materials and Methods

### 2.1. Study Population

This observational, prospective study included patients with acute intracranial bleeding treated at the Department of Anesthesiology and Intensive Care (ICU) at the tertiary care University Hospital between November 2015 and April 2017.

Inclusion criteria were: age ≥ 18 years, and ICU admission within 24 h after clinical diagnosis of acute intracranial bleeding confirmed by computed tomography (CT). The injuries included spontaneous subarachnoid hemorrhage, spontaneous intracerebral hemorrhage, and severe craniocerebral traumatic brain injury leading to epidural, subdural, subarachnoid, and intracerebral hematomas.

Exclusion criteria were: therapy with antiplatelet drugs (non-steroidal anti-inflammatory drugs, clopidogrel, prasugrel, ticagrelor, abciximab, tirofiban, eptifibatide) or anticoagulants (acenocoumarol, dabigatran, low-molecular-weight heparins), the presence of comorbidities such as thrombocytopenia (platelet count < 100,000/μL), and liver failure (bilirubin > 1.1 mg/dL). Patients whose estimated duration of treatment in the intensive care unit (ICU) was shorter than one day were also excluded from the study.

### 2.2. Ethics

The study protocol was approved by the Bioethics Committee of the Wroclaw Medical University (KB-391/2015) and complies with the Declaration of Helsinki of the World Medical Association. In all cases written informed consent was obtained from the patient or a legally authorized representative.

### 2.3. Outcome Assessment, Study Groups

The primary outcome of the study was mortality at ICU discharge. Patients were divided into two groups based on 28-day survival: Group 1—patients who survived and Group 2—patients who died.

### 2.4. Clinical Evaluation and Management

Patients were given treatment according to standardized management protocols (Carney et al. 2016 and Connolly et al. 2012) [19,20]. The APACHE II scale (Acute Physiology and Chronic Health Evaluation II) was used to assess the clinical condition of patients on admission to the ICU. Neurological status was assessed on admission to the ICU using the GCS (Glasgow Coma Scale) and the FOUR (Full Outline of Unresponsiveness) scale. The Rotterdam and Marshall scales were used to assess the severity of head injuries. Subarachnoid hemorrhages were classified on the basis of clinical symptoms according to the Hunt–Hess scale and the intensity of changes in the CT scan on the Fisher scale. Demographic data, medical history, and baseline clinical parameters were obtained shortly after admission.

### 2.5. Blood Sample Collection and Platelet Receptors Function Assessment

Blood samples for aggregometry were collected from the arterial line using a vacutainer system and 3 mL tubes with hirudin as an anticoagulant (Roche Diagnostics GmbH. Mannheim, Germany). Platelet function was assessed by whole-blood impedance aggregometry using the Multiplate analyzer (Roche Diagnostics. GmbH, Mannheim, Germany). For each patient blood samples were collected at the time of admission (day 1), and on days 2, 3, and 5. The Multiplate is a point-of-care analyzer and all tests were performed by one of two trained physicians immediately after collecting the blood in accordance with the manufacturer’s recommendations. The Multiplate analyzer has 5 separate channels allowing five tests to be carried out at the same time using specific activators of the platelet aggregation process: arachidonic acid (ASPI-test), adenosine diphosphate (ADP-test), collagen (COL-test), thrombin receptor activating protein 6 (TRAP-test), ristocetin (RISTO-test). Each test takes 6 min to complete. The system registers changes in electrical conductivity caused by the formation of platelet aggregates on each of the electrodes continuously for 6 min. The result is expressed as the mean of the measurements from the two electrodes as the area under the curve (AUC) and expressed in units (AU/min). Changes in electrical impedance are recorded for 6 min after activation. A quality control was run whenever reagents were reconstituted. The reference ranges for each test were provided by the manufacturer and are as follows: for the ADP-test 534–1220 AU/min, for the TRAP-test 941–1536 AU/min, for the RIST-test 896–2013 AU/min, for the ASPI-test 745–1361 AU/min, for the COL-test 459–166 AU/min. Blood samples for routine coagulation tests (PT, APTT, PLT, fibrinogen, and d-dimer) and blood count were collected at the same time-points.

### 2.6. Statistical Analysis

Continuous variables are presented as medians (interquartile range between the 25th and 75th percentiles); categorical data are presented as numbers and percentages. The distribution of the variables was not normal based on a Shapiro–Wilk test. A Mann–Whitney U test was used to assess differences between Non-survivors and Survivors for platelet aggregation and standard coagulation indices. The Kaplan–Meier curves and the log-rank test were used to assess differences in 28-day survival functions based on the global aggregometry analysis from the ADP-, COL-, TRAP-, RISTO-, and ASPI-tests. Univariate logistic regression analysis was performed to evaluate the association between baseline parameters and 28-day survival; the results were reported as odds ratio (OD) and 95% confidence intervals (CI). Categorical variables were analyzed using a chi-squared test. Significance was assumed if the probability of the null hypothesis was less than 5% (*p* ≤ 0.05). All of the analysis was performed on the 13.0 version of Statistica.

## 3. Results

### 3.1. Patient Clinical Characteristics

In the 20-month study period, 204 patients were admitted to the neurosurgical ICU. 130 patients admitted with a diagnosis other than acute intracranial bleeding were not included in the study. Of the 74 patients who were initially enrolled in the study, 34 were excluded due to the presence of one or more exclusion criteria. The final analysis was performed on 40 patients divided into two groups based on 28-day survival: Group 1—patients who survived (N = 29) and Group 2—patients who died (N = 11). The study groups were similar for age and gender. The flow diagram of the study is presented in Figure 1.

The severity of clinical and neurological status on admission to the ICU was assessed with clinical scores and Non-survivors were characterized by having a more severe clinical condition as indicated by worse clinical scores than Survivors (Table 1).

### 3.2. Treatment

In most of the patients (N = 25, 62.5%) multiple locations of hemorrhagic lesions were observed on the CT scan of the head performed on the day of diagnosis. A subarachnoid hemorrhage (SAH) was diagnosed in 26 patients; in 20 of them, the SAH was caused by a rupture of the cerebral vessel aneurysm (aSAH, Aneurysmal Subarachnoid Hemorrhage) and in 6 it was the result of trauma (tSAH, Traumatic Subarachnoid Hemorrhage); an extradural hemorrhage was diagnosed in 3 patients, a subdural hemorrhage in 14, an intracerebral hemorrhage in 15, and intraventricular hemorrhage in 14. In the group of 20 patients with aSAH, 13 patients were treated with intravascular embolization, 5 with aneurysm clipping, and 2 were treated conservatively. The remaining 20 patients were treated with a craniotomy with hematoma evacuation (9 cases) or with a decompressive craniectomy (8 cases); 3 patients were treated conservatively with monitoring the intracranial pressure (ICP). The choice of treatment method had no effect on mortality. Platelet aggregation in response to all the agonists used in the study was similar in patients with head injuries and in patients with ruptured aneurysms (*p* = 0.150).

### 3.3. Changes in Platelet Function

The platelet receptor response after stimulation with agonists such as arachidonic acid (ASPI), ADP, collagen (COL), TRAP, and ristocetin (RISTO) was significantly weaker in Non-survivors than in Survivors from day 1 to day 5. Platelet aggregometry results with the reference ranges for each test are shown in Figure 2. At baseline, the response of platelet receptors was below (TRAP-, RISTO-, ASPI-test) or at the border of the lower reference range (ADP-, COL-test) in Survivors, and below the reference range in all tests in Non-survivors, clearly indicating reduced platelet aggregation. In the following days, the median reactivity of platelet receptors remained reduced in Non-survivors, while in Survivors, platelet receptor reactivity increased, reaching normal values in the ASPI-, TRAP-, and COL-tests.

### 3.4. Analysis of Survival

The Kaplan–Meier estimator and the log-rank test were used to analyze the mortality rate associated with abnormal platelet aggregation (Figure 3). The 28-day survival rate in patients with and without aggregation disorders on the first day of ICU treatment was assessed. Global platelet aggregation disorders were diagnosed on the basis of values below the reference range in the ASPI-, ADP-, COL-, TRAP- and RISTO- tests. Patients with intracranial hemorrhage with platelet aggregation disturbances had a significantly lower survival rate compared to patients with normal platelet aggregation (log-rank test *p* = 0.014).

In addition, the results of the univariate logistic regression analysis between baseline d-dimer levels, aggregometry results, platelet count, and 28-day survival showed that aggregometry results within the reference range (OD = 7.200, 95% CI 1.468–35.318, *p* = 0.015) and d-dimers level (OD = 0.947, 95% CI 0.904–0.993, *p* = 0.023) were significant predictors of 28-day survival, while platelet count had no statistical significance (OD = 1.005, 95% CI 0.994–1.016, *p* = 0.343).

### 3.5. Other Parameters of Hemostasis

At baseline, there were no significant differences in clotting parameters between Survivors and Non-survivors, with the exception of d-dimer levels, which were higher in Non-survivors (2.3, IQR 1.3–6.4 vs. 19.9, IQR 9.4–45.8 mg/L, *p* = 0.003). D-dimer levels remained significantly higher in Non-survivors than in Survivors throughout the observation time. INR and APTT values were within the reference range at all times. The median platelet count was within the reference range in Survivors throughout the observation time (223, IQR 161–257; 208, IQR 128–235; 192, IQR 131–234; 206, IQR 160–254 × 10^3^/μL on days 1, 2, 3, and 5, respectively) and in Non-survivors within the reference range on days 1 and 2 (194, IQR 159–226; 169, IQR 130–195 ×10^3^/μL) and below on days 3 and 5 (88, IQR 72–167, 70, IQR 67–107 × 10^3^/μL, respectively). The differences in platelet count between Survivors and Non-survivors were statistically significant on day 3 (*p* = 0.001) and 5 (*p* = 0.006). The changes in the other standard coagulation parameters are shown in Table 2. Moreover, the changes in the level of c-reactive protein (CRP) and hemoglobin were analyzed. The median CRP values were elevated (Survivors: 10 mg/L, IQR 1–56; 72 mg/L, IQR 39–125; 127 mg/L, IQR 45–204, 55 mg/L, IQR 28-95 mg/L vs. Non-survivors: 2 mg/L, IQR 1–91; 124 mg/L, IQR 73–181; 195 mg/L, IQR 153–220; 170 mg/L, IQR 96–294 mg/L on day 1, 2, 3, and 5, respectively), but statistically significant differences between Survivors and Non-survivors were not observed until the 5th day (*p* = 0.019). The hemoglobin level was 12.9 g/dL (IQR 11.3–14.4), 11.7 g/dL (IQR 10.1–12.8), 11.0 g/dL (IQR 8.2–12.0), 11.0 g/dL (IQR 9.2–13.2) in Survivors and 12.1 g/dL (IQR 9.6–13.1), 11.1 g/dL (IQR 9.0–11.5), 10.4 g/dL (IQR 8.8–11.8), 8.5 g/dL (IQR 8.3–10.7) in Non-survivors on days 1, 2, 3, and 5, respectively; there were no significant differences between the groups (*p* > 0.05).

## 4. Discussion

Acute intracranial bleeding from head injuries and ruptured aneurysms are life-threatening conditions that appear suddenly and often lead to disability and even death. In this prospective study of blood coagulation disorders in patients with acute intracranial bleeding, a profound impairment in platelet receptor function was demonstrated. Despite normal platelet count on admission to the ICU, disturbances of platelet aggregation were observed in most patients. In Non-survivors, the median response to all tested agonists remained suppressed throughout the study period, while in Survivors it improved. Inhibition of the platelet activation was shown to be associated with patient mortality. The probability of 28-day survival was significantly lower in patients with aggregation disturbances identified on the first day of ICU treatment than in patients without disturbances (log-rank test *p* = 0.014).

Platelet dysfunction in patients with intracranial hemorrhage has been previously discussed in several studies. Different agonists were used to stimulate platelet receptors, and the results obtained were not consistent. Castellino et al. demonstrated that, in patients with an isolated TBI, there was a marked inhibition of platelet activation after stimulation with both ADP and AA agonists. The degree of inhibition of platelet activation after stimulation with ADP, but not with AA, was significantly higher in TBI patients with GCS ≤ 8 compared to those with a mild injury [14]. In a study by Ramsey et al., platelet dysfunction was evaluated using the thrombin receptor agonist peptide (TRAP) and ADP [21]. The authors confirmed that there was reduced ADP- and TRAP-mediated platelet aggregation, suggesting that the thrombin receptor pathway plays an important role in platelet dysfunction in head trauma. He et al. analyzed platelet dysfunction in aSAH patients on admission to the hospital and found a significant decrease in platelet reactivity in response to AA and ADP [17]; moreover, high systolic blood pressure and inhibition of platelet activation after stimulation with AA and ADP were independent risk factors for re-bleeding in the multivariate logistic regression model. Quite the opposite results were documented in a later study by Frontera et al., who found that mean platelet activation was within the reference range regardless of the severity of early brain injury after aSAH, as classified on the Hunt–Hess scale [16]. A possible explanation for this result could be the use of another measurement method; platelet activation was analyzed indirectly by thromboelastography and the maximum amplitude (MA) parameter. The MA value depends not only on the number and function of platelets but also on the level of fibrinogen, which is an acute-phase protein often elevated in patients after aSAH. In our study, in the group of patients with acute hemorrhagic damage to the CNS, the baseline platelet receptor response was below the reference range in the TRAP-, RISTO-, and ASPI-test and at the border of the reference range in the ADP- and COL-test in Survivors, and below the reference range in all tests in Non-survivors. These results clearly indicated decreased platelet aggregation. In the following days, the median reactivity of platelet receptors remained lower in Non-survivors but increased in Survivors, reaching normal values in the ASPI-, TRAP-, and COL-tests. Very early changes in platelet aggregation after intracerebral hemorrhage have been investigated in experimental settings. Using a large-animal model of combined TBI and hemorrhage, a significantly lower ADP-induced platelet aggregation was detected 15 min following injury that was further aggravated during the 2 h shock period. In our study, changes in platelet receptor activity were determined within 24 h of a clinical diagnosis of *acute hemorrhagic damage* to the CNS. Non-survivors showed a decrease in platelet response in ASPI-, ADP-, COL-, TRAP-, and RISTO-tests from admission to day 5, while Survivors showed a decrease on admission but normalization by day 5. In a recent study by Linblad et al., the changes in platelet receptor function over time after TBI were analyzed using multiple electrode aggregometry [22]. Most patients initially had ASPI test values below the reference range even though they were not receiving any treatment with COX inhibitors; the activity of the ASPI receptor increased in the following days of observation. In patients receiving cyclooxygenase inhibitors, low ASPI levels were reported, while platelet transfusion significantly improved platelet aggregation in the ASPI test. In another study by Perez et al. platelet aggregation was analyzed a few days after aSAH (5–10 days) and an increase in aggregation was observed after stimulation with ADP and arachidonic acid, suggesting an improvement in platelet function later in the ICU treatment, which is inconsistent with our results from the Survivors group [15].

We observed differences in platelet responses to the tested agonists between Survivors and Non-survivors on admission to the ICU and on subsequent days. A strong inhibition of platelet aggregation was observed initially in response to ristocetin, followed by normalization in Survivors, and an even stronger inhibition of platelet activation was found in Non-survivors. The mean platelet receptor response after stimulation with ADP and TRAP was significantly weaker in Non-survivors than in Survivors from days 1 to 5, while the receptor response after stimulation with arachidonic acid or collagen was significantly lower in Non-survivors only on day 1, and both pathways equilibrated on subsequent days. The exact role of platelet ligands such as vWF that facilitate the capture of platelets at the site of injury remains unknown. The platelet response to ristocetin depends on the von Willebrand factor (vWF) binding to the platelet GPIb receptor; thus, a decrease in the vWF or blockage of the GPIb receptors may be responsible for the low platelet aggregation in the RISTO test observed in both study groups on days 1 and 2 after injury. On days 3 and 5, there was an improvement in platelet aggregation in the Survivors and a significant deterioration in the Non-survivors. The mechanism that could potentially lead to the further reduction of platelet aggregation in the RISTO test in Non-survivors may be that the most of the circulating VWF was consumed in the process of binding to the platelets to form a clot. Our observation is consistent with the previously published results of Kornblith et al., showing significantly reduced platelet aggregation in response to ristocetin in patients with TBI compared to those without TBI [23]. The pathophysiological mechanisms leading to platelet dysfunction in patients with intracranial bleeding are not well understood and the evidence is mainly based on clinical observations. Tissue damage and hemorrhagic shock may be risk factors for platelet inhibition that occur immediately after trauma, since the release of ADP into the systemic circulation following tissue injury can lead to marked coagulation impairment [24]. However, the results of other studies based on an animal model indicated that TBI by itself might be sufficient to induce profound platelet dysfunction by a mechanism distinct from that which leads to the platelet dysfunction observed in trauma and shock. Castellino et al. suggested that the platelet dysfunction observed in the study that occurred immediately after the injury could be caused by a tissue factor released from the brain parenchyma following the mechanical disruption of the blood–brain barrier [14]. Better understanding of these mechanisms is needed to aid in the development of new diagnostic strategies and to enable better and faster identification of patients who may require treatment for coagulopathy.

Severe isolated TBI is not an independent risk factor for the development of coagulopathy; however, the incidence of coagulopathy in TBI is high and the numbers vary widely between studies, ranging from 7.7% to 63% [25]. The mortality rate in TBI patients who develop coagulopathy is high (17–86%) and depends on the severity of the TBI [25,26]. Despite numerous studies, early diagnosis of abnormalities in blood coagulation is a complex diagnostic problem in patients with intracranial bleeding after TBI or ruptured aneurysms. Routine coagulation tests, such as activated partial thromboplastin time, INR, and platelet count may be insufficient as they do not provide information on platelet function [27]. The results of standard blood coagulation tests may be within the reference ranges even when the overall blood hemostasis is abnormal [28]. In our study, the median platelet count measured on ICU admission was within the reference range in Survivors and Non-survivors and only aggregometric tests showed a profound impairment in platelet receptor function. This observation suggests that measuring platelet aggregation may be helpful in the early detection of bleeding disorders.

Predicting the mortality risk of ICU patients has many applications. It can be useful for planning resource allocation and evaluating the performance of intensive care wards. Accurately identifying patients with acute CNS hemorrhagic damage who are more likely to die and who can benefit most from additional monitoring or treatment remains a challenge. In this study, we found that routine coagulation tests are not sensitive indicators of platelet dysfunction in acute hemorrhagic CNS injury; there were no differences between the Survivors and Non-survivors in the results of routine coagulation tests such as platelet count, INR, aPTT, and fibrinogen level on ICU admission. The only difference noted between study groups was regarding d-dimers, which were 4× higher in Non-survivors than in Survivors, and this difference was observed up to day 5 of the study. Clinically significant dysfunction in platelet function with profound consequences for mortality was assessed using a Kaplan–Meyer survival analysis, which showed that patients with platelet aggregation abnormalities identified early on ICU admission had a significantly lower 28-day survival. The relationship between the inhibition of platelet aggregation and mortality in patients with hemorrhagic damage to the CNS as a result of head injuries and ruptured aneurysms has been studied by others, but the obtained results were inconclusive. Davies et al. demonstrated a correlation between early platelet ADP receptor inhibition and mortality in patients with isolated TBI [29]; however, the same relationship was not seen when arachidonic acid was used as a platelet receptor agonist. Kutcher et al. demonstrated that admission platelet hyporesponsiveness to TRAP, AA, and collagen were independent predictors of mortality in trauma patients [13].

The cause of death of patients in our study sample was the damage to the central nervous system. However, the influence of comorbidities on mortality cannot be excluded with certainty. The study group included patients with isolated brain injury; patients who required treatment with antiplatelet drugs or anticoagulants and patients with thrombocytopenia, and liver failure were excluded from the analysis. Another problem may be inherited platelet disorders, which are often only diagnosed after excessive bleeding. Presently, diagnostic capabilities have improved with next-generation sequencing methods. In a recently published study, Oved et al. identified the predicted loss of function mutations in genes related to platelets [30]. They found that 0.329% of the general population had a clinically significant predicted loss of variant function in the gene associated with platelets. As a result, these people are at risk of developing bleeding disorders, ranging from mild to severe. In our study cohort, none of the patients had a diagnosis of a genetic platelet disorder and none of the patients received treatment for this reason prior to ICU admission or during ICU treatment.

Although whole blood impedance aggregometry is accepted as a test for diagnosing platelet dysfunction, this test presents some specific problems. Indeed, the results can be influenced by various preanalytical conditions such as the type of anticoagulant, lipid plasma, hemolysis or a low platelet count. Based on the study by Hanke et al., the aggregometry results were reduced by 18.4% and 37.2% in blood samples with a platelet count of 100 and 50 × 10^3^/µL, respectively, compared with the results measured in blood samples with a platelet count within the normal range [31]. However, the authors emphasized that a large variation between individuals was observed, and some blood samples showed normal results even with a platelet count of 50,000 × 10^3^/µL. In a study of patients with traumatic brain injury treated in neuro-intensive care, the platelet count was positively correlated with the values of multiple electrode aggregometry (ASPI and TRAP test) [22]. However, there were other studies which did not confirm these results. In one study, the platelet count was not a significant predictor of platelet hypofunction after stimulation with ADP, TRAP, AA, and collagen in critically-injured trauma patients [13]. Similarly, Solomon et al. presented a retrospective study of impedance aggregometry responses to ADP, TRAP, and collagen in 163 trauma patients on admission to the ICU and found only a weak correlation between platelet count and agonist responsiveness [32]. The inconsistency between these studies indicates that there is a pressing need for future studies to overcome the problem of assessing inhibition of platelet activation in samples with low platelet counts.

This study has a number of limitations. Similar to other studies of platelet function in acute CNS hemorrhagic damage, our study is a preliminary, single-center experience. We recognize that the number of patients was low and the size of our cohort may have limited the identification of all the statistically significant changes in platelets, especially those related or not related to clinical course and mortality. Based on the study inclusion / exclusion criteria, patients with acute intracranial bleeding who had previously been treated with antiplatelet or anticoagulant drugs had to be excluded from the final analysis; therefore, the majority of patients with intracerebral hemorrhage were not included in the final analysis. The observed platelet inhibition, which occurred immediately after admission to the ICU, was the result of acute intracranial bleeding rather than prior treatment. In the authors’ opinion, this is a valuable observation, especially in the group with normothrombocytemia. In addition, blood samples were taken on admission to the ICU and continued on days 2, 3, and 5, which allowed for the interpretation of changes in platelet aggregation over time. Future studies regarding the effect of antiplatelet agents on the outcome of patients with acute CNS hemorrhagic damage are needed.

## 5. Conclusions

A profound inhibition of platelet activation indicating reduced platelet aggregation was demonstrated in patients with acute CNS hemorrhagic damage from head injuries and ruptured aneurysms. In Non-survivors, the platelet receptor response to all tested agonists remained suppressed throughout the study period, while it improved in Survivors.

Measurements of platelet function in patients with acute CNS hemorrhagic damage are strong predictors related to outcome. Bedside tests to measure platelet aggregation can be helpful in the early detection, diagnosis, and treatment of bleeding disorders.

## Figures and Tables

**Figure 1 jcm-10-02205-f001:**
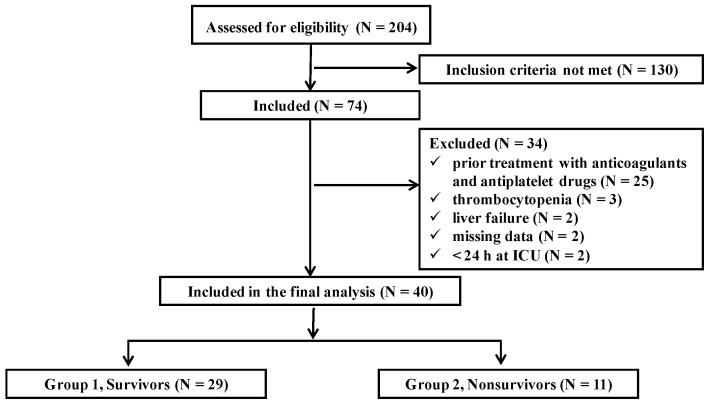
Flow diagram of the study. ICU, intensive care unit.

**Figure 2 jcm-10-02205-f002:**
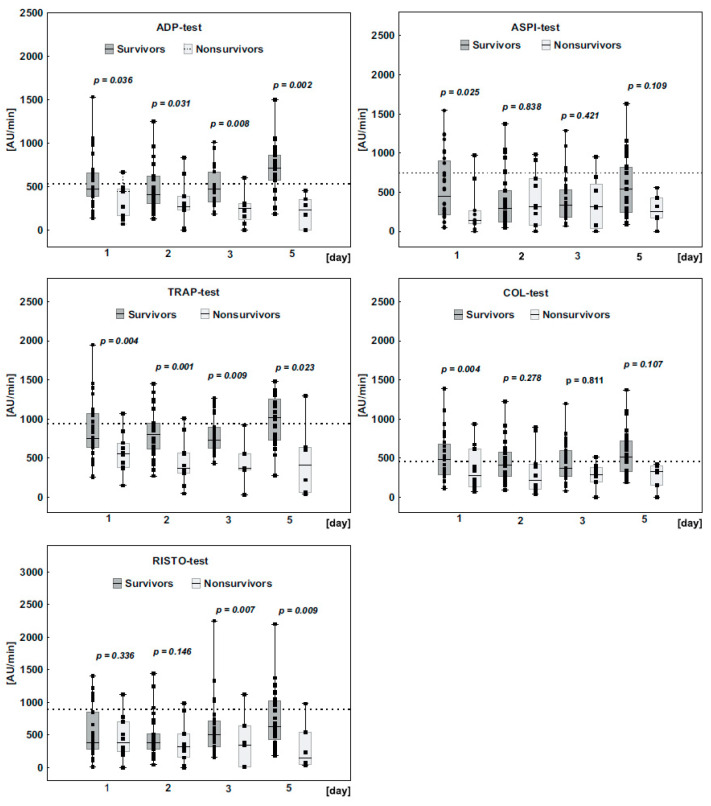
A comparison of the changes in platelet receptor activity between Survivors and Non-survivors in the following days of observation. A dotted line represents the lower reference range and the reference ranges for each test are as follows: ADP-test 534–1220 AU/min, TRAP-test 941–1536 AU/min, RISTO-test 896–2013 AU/min, ASPI-test 745–1361 AU/min, COL-test 459–1166 AU/min. The *p*-value represents statistically significant differences between the study groups. The box plots represent the median values (middle line) with upper and lower quartiles (box); the whiskers represent the minimum and maximum values.

**Figure 3 jcm-10-02205-f003:**
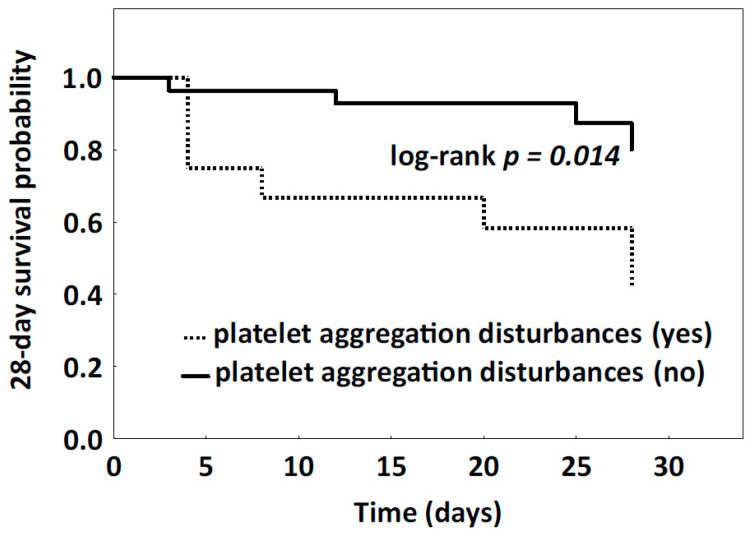
Probability of 28-day survival (Kaplan–Meier curve, log-rank test). Patients with platelet aggregation disturbances on the first day of ICU treatment are indicated by a dotted line and patients without aggregation disturbances by a solid line.

**Table 1 jcm-10-02205-t001:** Patient characteristics on admission to the ICU.

Parameter	All	Survivors	Non-Survivors	*p*
	N = 40	N = 29	N = 11	
Age (years)	54 (43–67)	53 (38–66)	64 (54–77)	0.064
Gender, male N (%)	23 (58)	15 (52)	9 (81)	0.082
GCS	7 (4–14)	13 (6–14)	4 (3–4)	<0.001
FOUR	8 (5–16)	16 (6–16)	3 (1–6)	0.001
APACHE	17 (12–27)	14 (10–18)	27 (25–29)	<0.001
Marshall score	5 (5–5)	5 (5–5)	5 (5–5)	0.500
Rotterdam score	4 (2–5)	3 (2–4)	5 (4–6)	0.012
Hunt-Hess score	3 (1–4)	2 (1–3)	5 (5–5)	0.008
Fisher score	3 (2–4)	3 (2–3)	4 (4–4)	0.033
ICU LOS (day)	10 (4–17)	11 (7–18)	5 (2–12)	0.085
Hospital LOS (day)	26 (16–50)	30 (23–67)	10 (4–25)	0.002

Values are presented as the median (interquartile range between the 25th and 75th percentiles) or as counts and fractions; *p*–value represents the difference between Survivors and Non-survivors; APACHE, Acute Physiology and Chronic Health Evaluation; FOUR, Full Outline of UnResponsiveness; GCS, Glasgow Coma Scale; ICU, Intensive Care Unit; LOS, length of stay.

**Table 2 jcm-10-02205-t002:** Coagulation parameters in Survivors and Non-survivors measured on days 1, 2, 3 and 5.

Day	Survivors	Non-survivors	*p*
	**INR**	**ref. range 0.9–1.3**	
**1**	1.05 (0.99–1.13)	1.10 (1.00–1.19)	0.531
**2**	1.08 (1.00–1.13)	1.17 (1.02–1.45)	0.036
**3**	1.03 (0.98–1.08)	1.16 (0.97–1.30)	0.019
**5**	1.00 (0.95–1.01)	1.20 (1.01–1.28)	0.019
	**aPTT**	**ref. range 25–37 sec.**	
**1**	28 (25–32)	29 (27–32)	0.610
**2**	29 (28–34)	34 (31–38)	0.235
**3**	31 (28–33)	34 (30–41)	0.045
**5**	30 (27–32)	32 (30–37)	0.042
	**Fibrynogen**	**ref. range 1.8–3.5 g/L**	
**1**	3.2 (2.6–3.4)	3.7 (2.8–4.5)	0.263
**2**	3.8 (3.3–5.0)	3.5 (2.6–4.2)	0.344
**3**	5.3 (3.4–5.8)	3.9 (3.1–4.8)	0.156
**5**	5.5 (4.3–6.7)	4.1 (2.5–5.8)	0.146
	**d-dimers**	**ref. range 0–0.5 mg/L**	
**1**	2.3 (1.3–6.4)	19.8 (9.4–45.8)	0.003
**2**	1.6 (1.0–3.4)	13.7 (11.7–30.9)	<0.001
**3**	2.3 (1.4–5.2)	26.5 (12.0–38.9)	0.014
**5**	1.8 (1.1–4.3)	10.9 (10.7–38.4)	0.002

Values are presented as the median (interquartile range between the 25th and 75th percentiles); *p*–value represents the difference between Survivors and Non-survivors; aPTT, activated partial thromboplastin time; INR, normalized ratio; ref. range, reference range.

## Data Availability

The data that support the findings of this study are available on request from the corresponding author Barbara Dragan. The data have not been made publicly available because they contain information that could compromise the privacy of the study participants.

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
