# Peer review of "Platelet Receptor Activity for Predicting Survival in Patients with Intracranial Bleeding"

_jcm, 2021, doi:10.3390/jcm10102205_

Round 1
Reviewer 1 Report
Coagulopathy is a common phenomenon in traumatic brain injury and a major contributor to complications including mortality. It has been a long standing knowledge that thrombocytopenia is a negative predictor for intracranial bleedings post-TBI, however, the mystery exists around bleeding tendencies in normothrombocytemic patients. Authors here investigate the role of platelet function testing for early diagnosis of the bleeding tendencies of these patients. Overall, this work has a good merit and suggests implementation of platelet aggs in TBI settings, which undoubtedly would result in better clinical outcomes in this patient cohort. However, there are a couple of points authors need to address:
- Authors (to some extent) replicated work done by Bellander's (2018) group however never mention it in this manuscript.
- Authors data support that coagulation likely does not play a role in coagulopathy post-TBI, however, don't discuss it much. This is very interesting to readers and this point of the study has to be more explicitly mentioned.
- Authors need to include a paragraph or two proposing their explanation for the findings described in this study. Why, authors think there is significant decrease in plt aggs, in non survivors for all the agonist, except risto at Day 1. Overtime risto becomes different between groups, whereas GPVI and COX pathways equilibrate. This is a of a great interest and would undoubtedly enhance the manuscript.
- Please seek professional English editing. This will improve readability of the manuscript.
Reviewer 2 Report
The authors present a research article about platelet dysfunction as a predictive factor of outcome in patients with intracranial hemorrhage.
Blood coagulation disorders are frequent in patients with intracranial bleeding, and the role of platelet function is an interesting and actual topic, still much debated.
The manuscript is well written and interesting to read. The statistical analysis is good performed and the results support the conclusions. In addition, the discussion is well-argued and stimulating.
Congratulations to the authors.
Reviewer 3 Report
The study by Dragan et al. examines platelet dysfunction as a predictor of survival in patients with intracranial hemorrhage. For this purpose, they tested platelet aggregation, induced by 5 major platelet agonists, using a multiplate analyzer.
This study is part of a diverse and often contradictory literature, but it also presents weaknesses
Comments
Line 58 : "The 158 study groups were similar for age and gender". However, Table 1 shows a difference between groups (in terms of % male and age patients)
In addition, the medians (instead of mean) would probably be more informative
- Figure 2: please replace bars with boxes/dot plots showing individual values in the aim to evaluate value distribution
- The fact that D-Dimers continue to be significantly elevated in non-survivors is in favor of a coagulopathy. Did the authors test the predictive value of DDi level on survival to compare with that of the aggregation results?
- In non-survivors platelet count was low or very low on days 3 and 5, meaning unreliable most results of aggregations. Could you provide the performances of Multiplate analyzer according to platelet count? Again, what was the predictive value of platelet count on mortality, alone or in combination with DDi, compared to aggregation results?
• Could you provide:
- the haemoglobin level in patients? To interpret Multiplate results
-CRP value to interpret fibrinogen levels with regard to inflammatory reaction
-Hemolysis markers to interpret a potential desensitization of ADP receptors secondary to ADP release from red blood cells
The term " inhibition of receptor " is regularly used but this is inadequate. Indeed, this study does not allow to show this, especially since all agonists seem to induce a reduced activation which would suggest rather a global defect (at the level of the receptor of aggregation, the αIIbß3 integrin, or at the level of the signaling, unless it results from an external factor). All the above caveats aside, the term that corresponds to their results is "inhibition of platelet activation" or “decrease in platelet reactivity”
Reviewer 4 Report
This manuscript is a beautiful study by Dragan et al, where the authors showed that platelet receptor activity can be a good parameter for predicting survival in patients with intracranial bleeding. The manuscript possesses a very simple study design by following patients admitted in the ICU and then measuring their platelet activity. This outcome of the study is important for diagnostic purpose and hence this could be a nice addition to MDPI-JCM. However, I have two minor concerns at this stage which are summarized below. If the other reviews are favorable, I would like the authors to discuss this in the discussion section before this can be published.
Minor Comments:
(1) I think the number of patients studied in this manuscript is very low (survivor = 29 and non-survivor = 11). And looking at the significance values between these groups for many comparisons are not really striking. Hence, the observation is really important, but statistically, I am still not very sure how strong they are. I would like to see if the authors can somehow improve this or have an explanation for this.
(2) Many platelet disorders are associated with genetic variations in human population. The authors took patients who came randomly to the ICU. They are expected to have different genetic make ups. How do they normalize for this issue? Additionally, the survival of patients may depend on the depth of the injury, if they had injury at any other body parts and also on physiological/other disease conditions that they might have before the injury. Hence, in these cases, the death may not be related to platelet aggregation parameter solely. How can the authors explain that?
